# Inelastic Neutron Scattering Study of Phonon Dispersion Relation in Higher Manganese Silicides

Randy Belanger [1], James Patrick Clancy [1], Sheetal Jain [1], Zahra Yamani [2], Yu-Chih Tseng [3] and Young-June Kim [1,*]

1   Department of Physics, University of Toronto, Toronto, ON M5S 1A7, Canada
2   National Research Council, Canadian Neutron Beam Centre, Chalk River, ON K0J 1J0, Canada
3   CanmetMATERIALS, Natural Resources Canada, Hamilton, ON L8P 0A5, Canada
*   Correspondence: youngjune.kim@utoronto.ca

**Abstract:** We report inelastic neutron scattering (INS) measurements of the phonon dispersion relation in higher manganese silicides (HMSs). A large ingot of HMS is synthesized using a slow cooling method, which is found to have $Mn_{15}Si_{26}$ as the primary phase. The sample is composed of highly oriented crystallites as confirmed by a neutron pole-figure study and thermal conductivity data. Our INS results are mostly consistent with earlier experimental and theoretical phonon studies in HMS, including the presence of a low-lying twisting mode. However, some discrepancies are also observed. Most notably, a 5 meV gap at the zone center and the softer dispersion relation of the low-lying twisting mode. We discuss the potential origins of these observations and their implications for the thermal properties of HMS.

**Keywords:** inelastic neutron scattering; thermoelectric materials; higher manganese silicide; phonon dispersion relation

## 1. Introduction

Higher manganese silicides (HMSs) $Mn_nSi_{2n-m}$ are an example of a Nowotny chimney ladder phase, whose average structure consists of a commensurate number of tetragonal Mn chimney sublattices and Si ladder sublattices [1,2]. All phases of HMS have a tetragonal crystal structure with similar *a* lattice parameter but with significantly different *c* lattice parameters due to incommensurate lattice spacing between Si and Mn sublattices. This results in various competing phases with slightly different compositions: $MnSi_{1.75}$, $MnSi_{1.727}$, $MnSi_{1.733}$, and $MnSi_{1.741}$. From a structural point of view, these phases are better described with large unit cells: $Mn_4Si_7$, $Mn_{11}Si_{19}$, $Mn_{15}Si_{26}$, and $Mn_{27}Si_{47}$, respectively. The local structure is identical for these different phases, but the c lattice parameters differ substantially, ranging from about 17 Å for $Mn_4Si_7$ to about 118 Å for $Mn_{27}Si_{47}$. Because of the structural similarity, these phases tend to coexist in the same sample. Such inherent structural disorder makes them a perfect candidate material for thermoelectric applications, which require electron crystal/phonon glass structures [3]. For a recent review of HMS thermoelectric materials, see Refs. [4,5]

Currently, there is active research underway to bring thermoelectric materials into commercialization for applications where significant waste heat could be converted back into usable electricity [6–12]. The performance of thermoelectrics is usually judged based on the thermoelectric figure of merit $ZT = S^2 T / \kappa \rho$, where $S$ is the Seebeck coefficient, $T$ is the temperature, $\rho$ is the resistivity, and $\kappa$ is the thermal conductivity. The thermal conductivity can be further subdivided into lattice and electron components as $\kappa = \kappa_e + \kappa_l$. The $ZT$ in pure HMS was found to reach a maximum of about 0.4–0.5 at around 700–900 K [13–16] and up to around 0.82 in HMS doped with Ge, Re, or Al [15–19] (a benchmark value is $ZT \sim 1$). Unfortunately, $S$, $\rho$, and $\kappa_e$ are all inter-correlated, electron-dependent quantities, and thus it is very difficult to improve $ZT$ by improving any one of these quantities, as it negatively

impacts the other two [6]. Thus, most researchers focus on reducing the lattice thermal conductivity. It is already known that HMS has a low and anisotropic ($\kappa_{\parallel c} > \kappa_{\perp c}$) lattice thermal conductivity [20,21]. However, in order to reduce the lattice thermal conductivity further, it is useful to understand the lattice dynamics, and for this, knowing the phonon dispersion is indispensable.

In their combined study of density functional theory (DFT) and inelastic neutron scattering (INS) on a mostly $Mn_{27}Si_{47}$ phase sample, Chen et al. [22] provided a picture of the phonon dispersion in HMS. Of note, they showed the existence of a low-lying phonon mode, which their calculations claim to be an optical mode caused by twisting motions of the Si helices. This mode is believed to have strong anisotropy between the *a*- and *c*-axis directions. While the existence of such a twisting mode was verified in their INS results, the lack of a detailed dispersion relation study makes it difficult to confirm some of the theoretical predictions, such as the dispersion along the c-axis and the existence of a 1 meV gap at the zone center for the twisting mode. Furthermore, according to their DFT calculations, the average sound velocities of the major heat-carrying modes (typically acoustic) are higher along the *c*-direction, which is opposite to the expectation based on the lattice thermal conductivity anisotropy, which is higher along the *a*-direction. They give possible reasons to explain the lattice thermal conductivity anisotropy, including the reversal of the group velocity anisotropy at high energies, increased phonon scattering, and lower diffusion coefficient along c. In addition, increased boundary scattering of phonons along the *c*-axis due to the stacking of different HMS phases or MnSi layers could overshadow the sound velocity anisotropy.

Here, we report a comprehensive INS study of the HMS phase $Mn_{15}Si_{26}$. This study was made possible by a very large HMS sample obtained using a simple cooling method. We confirmed that this sample is made up of highly oriented crystallites, therefore allowing us to obtain the phonon dispersion relation. We note that oriented polycrystalline samples were prepared in the past by sintering under uniaxial pressure [23]. We were able to confirm the dispersion of all of the acoustic phonons, as well as the 'twisting' mode, propagating along the *a*-axis, which deviates slightly from the calculation reported in Ref. [22]. Furthermore, we observed multiple acoustic phonon modes propagating along the *c*-axis that span multiple Brillouin zones, whose dispersion relation agrees well with the earlier calculation result. Overall, our INS results, while being consistent with the twisting mode scenario proposed in Ref. [22], reveal some interesting discrepancies that require further study.

## 2. Experimental Details

### 2.1. Material Synthesis and Characterization

A roughly 300 g ingot of HMS was synthesized at Natural Resources Canada's CanmetMATERIALS using a solid state reaction between high purity (99.999%) Si and (99.9%) Mn powders in an inert environment. The Si and Mn powders were mixed in a ratio of 46.75 wt% to 53.25 wt% Si:Mn, heated above the melting point of Si, and then subsequently cooled back to room temperature over a period of 24 h. From the 300 g ingot, a roughly 100 g slice was cut (radius ∼2.5 cm; thickness ∼1 cm), which was further subdivided into four pieces (see Figure 1). Piece A was ∼50 g, Piece B was ∼25 g, and Piece C and Piece D were another ∼25 g total.

Crystal structure determination was performed with a Bruker (Billerica, MA, USA) D8 Advance X-ray powder diffractometer using Cu $K_\alpha$ radiation and analyzed using the software GSAS-II [24]. The crystal structure was investigated by grinding up a small amount of piece C and performing powder diffraction. Thermal transport measurements were performed with a quantum design physical property measurement system (PPMS) for the temperature range 2–300 K under high vacuum. A small bar ($3 \times 3 \times 10$ mm) cut from the oriented Piece B was used in the thermal transport measurements.

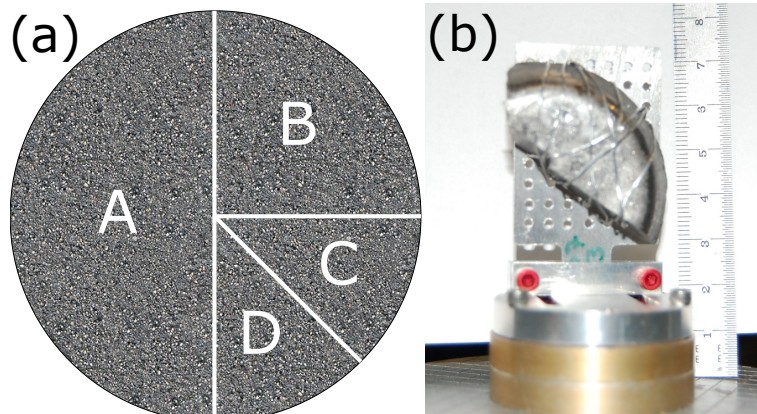

**Figure 1.** (**a**) Relative sizes of the various pieces of single crystalline Mn₁₅Si₂₆. Piece labels, A–D, are described in the text. The total mass is ∼100 g with a radius of ∼2.5 cm and thickness ∼1 cm. (**b**) Image of Piece A.

*2.2. TEM*

The transmission electron microscopy study was conducted using a FEI (Hillsboro, OR, USA) Tecnai Osiris system equipped with an X-FEG gun at 200 keV. In the scanning TEM (STEM) mode using a bright field, high-angle annular dark field (HAADF) detectors were used in combination with energy-dispersive spectroscopy (EDS). For the TEM study, samples were prepared using focused ion-beam (FIB) lift out.

*2.3. Neutron Scattering*

Piece A of the HMS crystal was aligned using the E3 spectrometer at the National Research Universal (NRU) reactor at Chalk River Laboratories to determine crystal quality and to align the crystal in the ($HK0$) and ($H0L$) scattering planes.

Inelastic neutron scattering measurements were carried out using the C5 polarized beam triple-axis spectrometer (also at NRU) with a vertically focused pyrolytic graphite (PG) monochromator and analyzer, using the (0, 0, 2) reflection at a fixed $E_f$ = 13.7 meV, while the incident energy was scanned at each $q$ point. Two PG filters were used in the scattered side (after sample) to reduce the background, and beam collimations were set (none, 0.8°, 0.85°, and 2.4°) in front of the monochromator, sample, analyzer, and detector, respectively. The sample was mounted within a closed cycle refrigerator (ICE Oxford Witney, UK) and cooled to 200 K, unless otherwise indicated.

**3. Results**

*3.1. Structure Determination, Crystal Quality, and Thermal Transport*

In Figure 2a, the Rietveld refinement result of the powder X-ray diffraction data are plotted. The refinement was carried out for the Mn₁₅Si₂₆ phase with lattice constants $a = 5.525(3)$Å and $c = 65.466(6)$Å, in $I\bar{4}2d$ space group. The fit results were $R_{wp} = 9.11\%$ with reduced $\chi^2 = 2.772$. The phase purity was determined to be 94.5(5)%. We tried refinements using multiple phases up to five phases, but the refinement result did not improve over the single-phase refinement presented here.

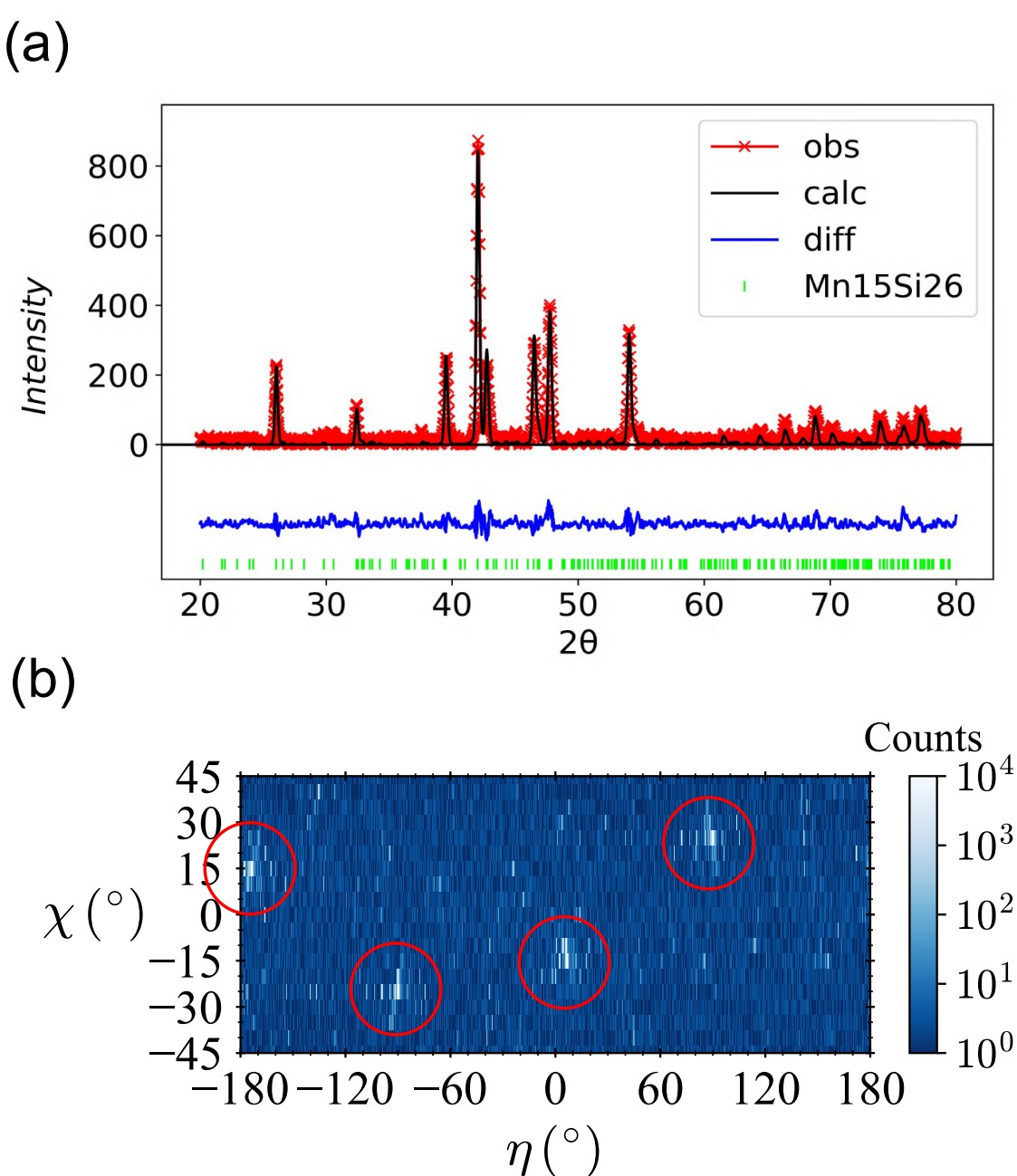

**Figure 2.** (**a**) Rietveld refinement of powdered HMS sample showing $> 90\%$ $Mn_{15}Si_{26}$. The powder X-ray diffraction data are plotted in red symbols. The black line is the fit and the blue line is the difference between the data and the fit. The green tick marks represent peak positions. (**b**) Check of the crystal quality of the HMS sample using neutron diffraction confirms 87% of the ~50 g sample is composed of a single grain. The circles denote four symmetry-related $(2, 0, 0)$ peaks: $(\pm 2, 0, 0)$ and $(0, \pm 2, 0)$.

Further neutron pole-figure measurements were performed to check the crystallinity of Piece A. Since neutrons interact weakly with materials, we can use them to probe the large-scale bulk crystallinity of the sample. As is shown in Figure 2b, we used a neutron spectrometer to map out a region of reciprocal space, at a fixed detector position (i.e., fixed $2\theta$), looking for the (200) peak and its symmetry equivalents. The two angles $\eta$ and $\chi$ together map out every possible orientation in which the crystal can be aligned. If all possible orientations are considered as lying on a unit sphere, the angle $\chi$ can be thought of

as the polar angle (measured from the equator) and $\eta$ as the azimuthal angle. Here, we scan $\chi$ in 5° intervals from $-50°$ to $50°$, and scan $\eta$ a full 360° at each $\chi$, which encompasses roughly 75% of the reciprocal space for a constant $2\theta$, which is set for the (200) reflection. Due to the 4-fold rotational symmetry of the tetragonal $Mn_{15}Si_{26}$ unit cell, we expect to see four (200) reflections each separated by 90° on the unit sphere. If there are multiple grains, we would expect to see greater than four reflections. As can be seen from the bright regions in Figure 2b, there are only four main reflections visible. We are assured that there are no reflections from other grains outside of our region of interest due to the 4-fold rotational symmetry (if there was one reflection located outside our region, then there would need to be one 90° from that, which must be back inside our region). Furthermore, we can determine angle $\theta$ between each peak by using the relation for the angle between two points on a sphere:

$$\cos\theta = \cos\chi_1 \cos\chi_2 \cos(\eta_1 - \eta_2) + \sin\chi_1 \sin\chi_2. \tag{1}$$

Using this, we determine the average angle between each of the peaks to be $90.0 \pm 1.5°$, as expected. By analyzing the intensity of the four major peaks and comparing them to the intensity of the remaining reciprocal space, we also estimate that the most dominant grain comprises >87% of the volume of the sample, which roughly matches the phase fraction of $Mn_{15}Si_{26}$. There also appear to be other smaller grains making up the majority of the remaining volume only a few degrees away from the main grain.

Figure 3 shows the thermal conductivity of small bars ($3 \times 3 \times 10$ mm) cut from the oriented Piece B. The thermal conductivity along the direction perpendicular to the c-axis is about a factor of 2–3 times larger than that along the direction parallel to the c-axis, similar to previously reported results [20,22]. In their study, Chen et al. [22] proposed that the anisotropic thermal conductivity is due to various factors, such as increased phonon scattering and lower diffusion constant along the c-direction. An important factor is the structural disorder along the c-direction in this type of structure, in which multiple competing phases coexist, increasing phonon scattering due to the structural disorder.

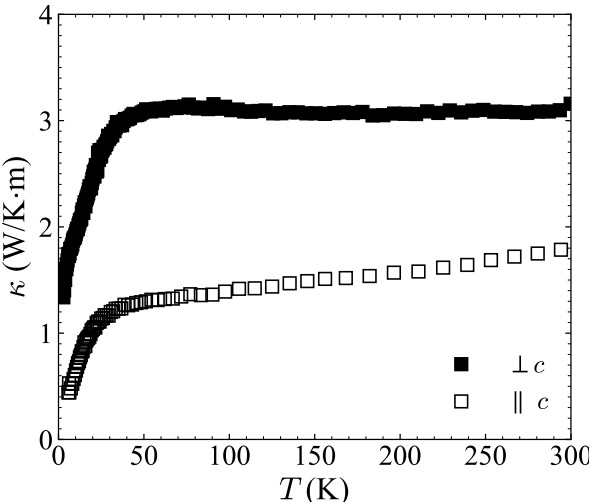

**Figure 3.** Total thermal conductivity along directions parallel and perpendicular to the c-axis of $Mn_{15}Si_{26}$. The measurement error bars are smaller than the symbol.

### 3.2. TEM

A wide area scanning transmission electron microscopy (STEM) image of the HMS sample is shown in Figure 4a. In this image, we can clearly see a small amount of dark precipitates, which are Mn-rich phases, such as MnSi. The MnSi striations running perpendicular to the c-axis have been discussed extensively in earlier studies [23,25]. Also clear from this image is that the sample is composed of grains of about 100 nm, suggesting that

the crystal can be considered a highly oriented mosaic crystal. Note that a collection of well-aligned nano-scale grains is fully consistent with the neutron pole figure results (which are sensitive to average crystal orientation over the bulk of the sample). The angular width of the peaks in the neutron pole figure implies that these nano-domains must be aligned to within about 5 degrees. This is in agreement with the inference from low-temperature thermal conductivity measurements as will be discussed in Section 4. The zoomed-in TEM image of the white box is shown in Figure 4b and illustrates the defect structure in more detail. These types of defects were examined in depth previously by Ye et. al. [25], and were attributed to dislocations in the silicon sublattice of the HMS. Such defects are sources of contrast in TEM imaging, and the wide area image shown in Figure 4a shows that such defects are abundant within the HMS ingot. Locally near the defect, there is a slight change in the Mn:Si ratio caused by dislocations, such as the addition of extra atomic planes in the Si sublattice. This observation may explain the presence of other phases of HMS, namely $Mn_{24}Si_{47}$ and $Mn_4Si_7$ as revealed in the powder XRD data.

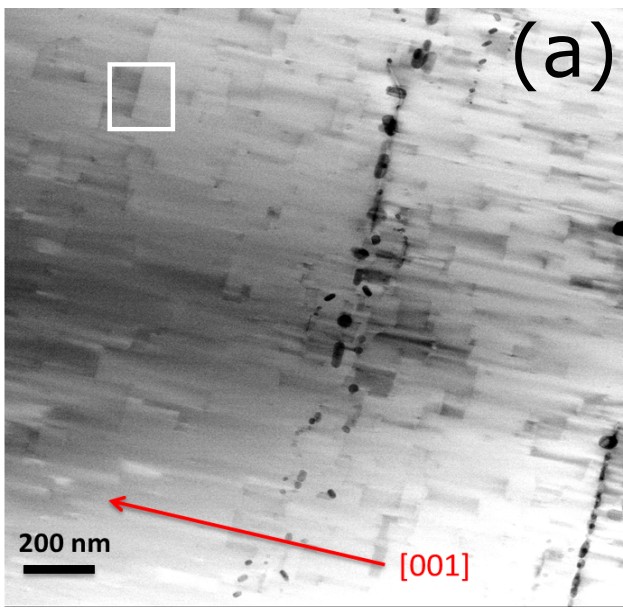

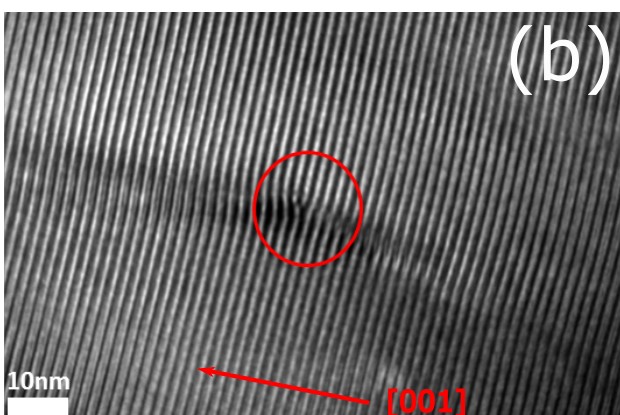

**Figure 4.** (**a**) STEM imaging of a wide area of the HMS. The dark precipitates are Mn-rich phases, such as MnSi. The features with apparent alignment along the c-axis (an example in the white box) are caused by defects such as those shown in part b. (**b**) High-resolution TEM imaging of an apparent "edge dislocation" in the HMS sample. The bright lines correspond to the part of the HMS cell where Mn and Si atoms are located closely together. Extra planes in the Si sublattice are responsible for the overall appearance of an edge dislocation.

### 3.3. Inelastic Neutron Scattering

The intensity, $I$, of the phonon peak in an INS measurement is proportional to $I \propto (\mathbf{Q} \cdot \xi_{js})^2$, where $\mathbf{Q}$ is defined as $\mathbf{Q} = \mathbf{G} + \mathbf{q}$; $\mathbf{G}$ being the position of a reciprocal lattice point and $\mathbf{q}$ a reciprocal lattice vector which points in the direction of phonon propagation; $\xi_{js}$ is the polarization of phonon mode $s$. This relation is useful for determining where to look for certain phonon modes and in assigning phonon mode polarization. For example, the intensity for a longitudinal acoustic (LA) mode along the c-axis would be greatest for $\mathbf{Q}$ pointing along the $(0, 0, L)$ direction with large $L$. The transverse acoustic (TA) mode is invisible in this direction because $(\mathbf{Q} \cdot \xi_{js}) = 0$. On the other hand, to measure the TA mode propagating along the c-axis, one should study the dispersion relation along $(2, 0, q)$ with small $q$ (see Figure 5). Furthermore, for acoustic phonons, $\lim_{q \to 0} I \propto F_N(G)$, where $F_N$ is the nuclear structure factor at a Bragg peak, G. Thus, acoustic phonon modes are also more intense near intense Bragg peaks, such as those at (200) and (220) in HMS.

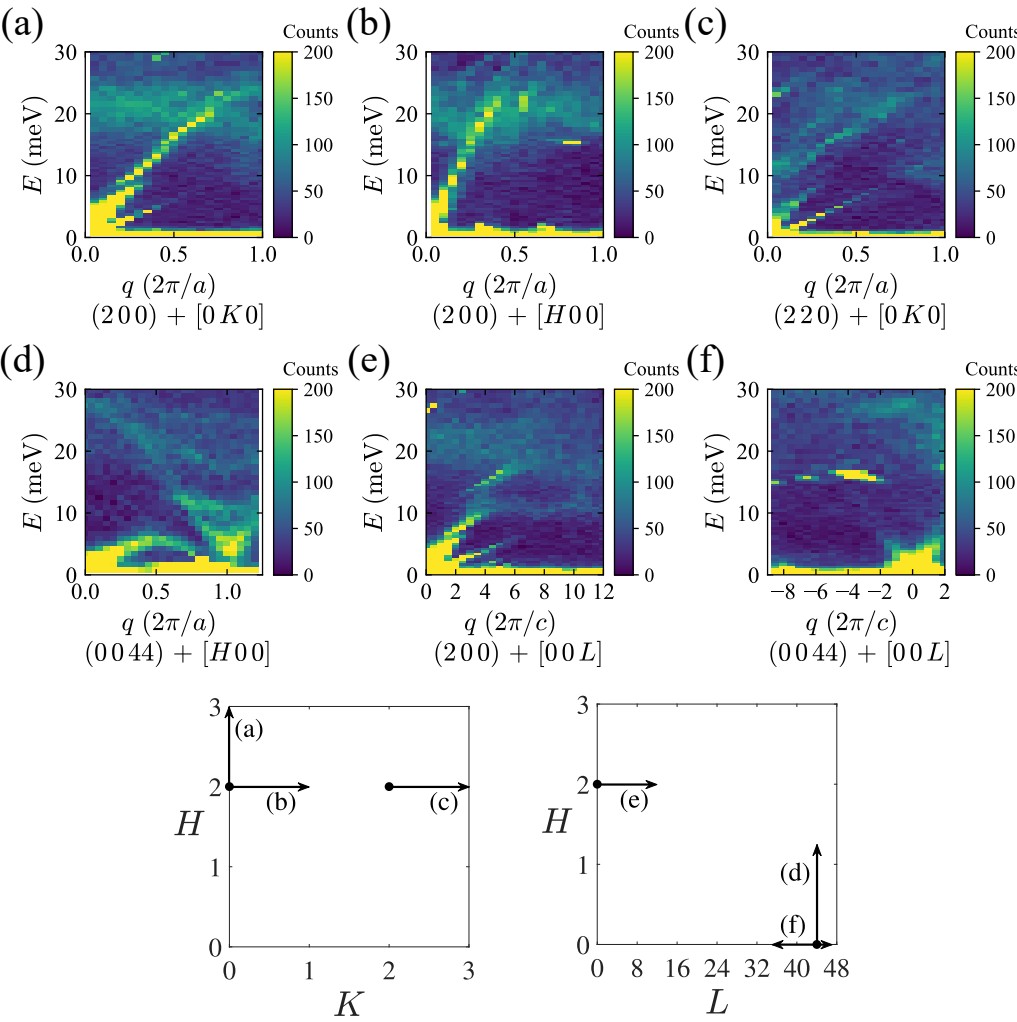

**Figure 5.** INS intensity for the $Mn_{15}Si_{26}$ crystal at 200 K: (**a**) along $[0K0]$ in the (200) Brillouin zone, which is most sensitive to ab-polarized transverse modes; (**b**) along $[H00]$ in the (200) zone, which is most sensitive to longitudinal modes; (**c**) along $[0K0]$ in the (220) zone, which is sensitive to longitudinal and transverse modes; (**d**) along $[H00]$ in the (0 0 44) zone, which is most sensitive to c-polarized transverse modes; (**e**) along $[00L]$ in the (200) zone, which is most sensitive to ab-polarized transverse modes; and (**f**) along $[00L]$ in the (0 0 44) zone, which is most sensitive to longitudinal modes. The scan directions for each panel are shown in the reciprocal lattice planes shown at the bottom.

Each color plot in Figure 5 is a combination of a series of line scans of the INS intensity as a function of energy obtained at each value of $q$. The sample was first oriented such that the $(HK0)$ zone is in the scattering plane in order to look for longitudinal and transverse modes polarized and propagating in the ab plane (Figure 5a–c). Note that (a)–(c) were obtained in the $(HK0)$ scattering plane, while (d)–(f) were obtained in the $(H0L)$ scattering plane. The scan directions are shown in the $(HK0)$ and $(H0L)$ planes at the bottom of Figure 5.

In Figure 5a, the triple-axis spectrometer is most sensitive to transverse modes, and we notice two well-defined modes. The acoustic mode at higher energy is the ab-polarized TA mode, which hybridizes and undergoes an avoided crossing with an optical mode at about $q \sim 0.2$ and $E \sim 7$ meV. The lower-lying branch that extends to about 5 meV at $q \sim 0.5$ is probably due to Bragg tails, which are often found near a strong Bragg peak and is an experimental artifact [26]. This assignment is based on the fact that similar linear branches with the same slope are observed along the c-direction (see panel (e)) and also near the (2 2 0) Bragg peak shown in panel (c). The latter branch seems to extend well past the zone boundary. Another piece of evidence that confirms the origin of the artifact is the temperature dependence (shown in Supplementary Figure S1). While phonon intensity follows the thermal factor (Bose–Einstein distribution), Bragg tails are temperature-independent. In Figure 5b, there is one well-defined mode that extends to about 20 meV, which is assigned to the LA mode. There is also a wide band of optical phonons that are present in both Figure 5a,b between about 15 and 25 meV.

In Figure 5c, we switch from the (200) zone to the (220) zone. Here, we should be equally sensitive to both the LA and ab-TA modes but with less intensity than in Figure 5a,b. Indeed, both the ab-TA and LA branches are observed between 10 and 20 meV, as well as below 5 meV, where these two modes are indistinguishable. Between 7 meV and 10 meV, the avoided crossing is also observed (similar to panel (a)). However, we also observe a fairly well-defined optical mode near the zone center and around 10 meV.

In Figure 5d–f, we show the results of the INS obtained in the $(H0L)$ scattering plane in order to look for phonon modes polarized and propagating in the a–c plane. The INS data shown in Figure 5d are very intriguing. For the orientation of the sample in this panel, we are most sensitive to the c-polarized transverse modes propagating along the a-axis. First, we can clearly observe the full dispersion of the low-lying twisting mode from $q = 0$ to $q = 1$ with a maximum of about 5 meV. This mode is consistent with the observation of the twisting mode reported in Ref. [22]. What is interesting is the fact that this twisting mode intensity seems to be decreasing as it approaches (1 0 44), while the other optical modes have strong intensity near (1 0 44). We note that while (1 0 44) is not an allowed Bragg peak, (1 0 45) is a strong Bragg peak. Since the c-lattice constant is very long, (1 0 44) and (1 0 45) are very close (only 2% difference in Q), and our measurement integrates over a fair amount of L-ranges. The observed intensity change of the twisting mode could indicate that the twisting mode is strongly polarized along the c-axis. Secondly, we observe a couple of interesting optical modes near (1 0 44) – probably (1 0 45) as discussed above. One of these modes have a gap of about 5 meV and disperses steeply up to 15 to 20 meV, while another mode with a gap of about 10 meV seems to show much weaker dispersion and crosses the first mode, around $q = 0.7$.

Figure 5e shows the dispersion of the ab-TA mode along the c-axis. This should be compared to panels (a) and (c). Here, the lowest-lying branch is due to the Bragg tails as discussed above. Overall, dispersion looks quite similar to the data shown in panel (a). The big difference is the clear avoided crossing behavior near 10 meV and the clearly resolved optical dispersion around this energy between $q = 4$ and $q = 12$. We note that the phonon dispersion relation in quasi-two-dimensional materials often appears to ignore the true reciprocal lattice periodicity, but this is due to the strong modulation of the phonon intensity from one Brillouin zone to another. The underlying physics is that the propagation of short-wavelength phonons tend to be dominated by local arrangements rather than the periodicity. Finally, Figure 5f shows the presence of weak dispersion near the zone center

up to 5 meV. Along this direction, the intensity drops very quickly away from the zone center, making it difficult to obtain quantitative information. The pixels showing strong intensity around 15 meV and $q = -4$ may also be an artifact.

In the earlier study by Chen et al. [22], a phonon dispersion relation calculated with DFT was reported, which seemed to describe the observed INS data well. In Figure 6, these calculation results are reproduced with our INS data overlaid. Here, we plot the peak positions from the Gaussian fitting of the INS intensity as a function of energy for each $q$-value. We note that the DFT calculation was performed for $Mn_4Si_7$, a similar HMS phase to $Mn_{15}Si_{26}$, but with a c-axis about 3.75 times shorter than $Mn_{15}Si_{26}$. The notation $\Gamma - Z$ in the plot refers to the $Mn_4Si_7$ unit cell used in the calculation. The overlaid data along this direction are plotted according to the correct physical unit. In other words, the purpose of the scaling is to compare the calculated and experimental results corresponding to the same $q$ in units of $\text{Å}^{-1}$.

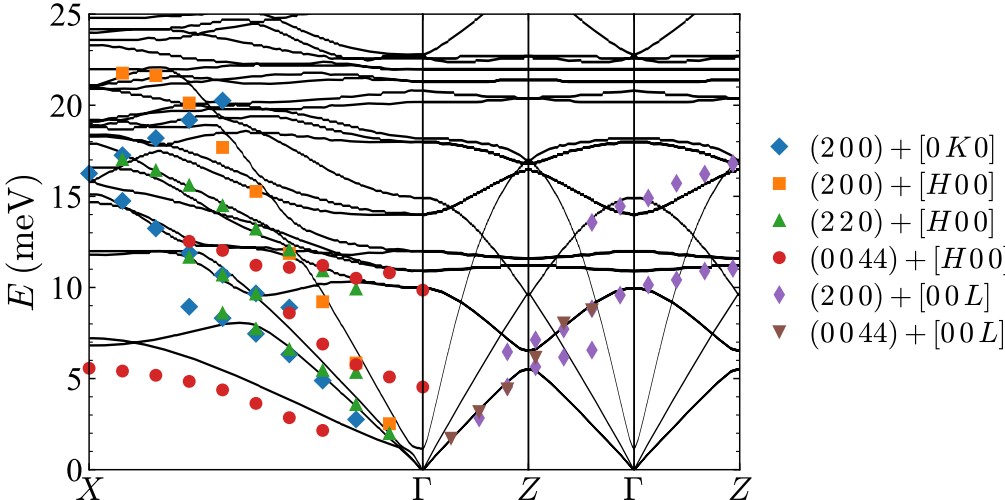

**Figure 6.** Experimental phonon dispersion of $Mn_{15}Si_{26}$ overlaid with the calculated harmonic dispersions of $Mn_4Si_7$ from Ref. [22] (solid black lines) in absolute reciprocal space units. Each set of markers represents a Gaussian fitting of the peak position of the INS intensity as a function of energy for each of the plots in Figure 5; $(200) + [0K0]$ (blue diamonds; Figure 5a), $(200) + [H00]$ (orange squares; Figure 5b), $(220) + [0K0]$ (up green triangles, Figure 5c), $(0044) + [H00]$ (red circles; Figure 5d), $(200) + [00L]$ (thin purple diamonds; Figure 5e), $(0044) + [00L]$ (down brown triangles; Figure 5f).

Note that not all phonon modes can be reliably identified in our experiments. Nonetheless, the overall agreement between the phonon dispersion data from our measurements and the calculation results is excellent. There seem to be only two meaningful differences, both based on the data shown in Figure 5d along the $\Gamma \rightarrow X$ direction. First, our twisting mode data systematically undershoot the calculation. Second, the c-TA mode that we measure does not continue down to the zone center as shown in the calculation but instead has a gap of about 5 meV (red circles near $\Gamma$). These modes will be further discussed below.

## 4. Discussion

While the thermal conductivity for both samples in Figure 3 agrees with the literature at room temperature, the low temperature (<100 K) region is markedly different. Most results in the literature show a peak in the thermal conductivity in this low-temperature region [16,20,22], while ours do not. This peak essentially results from the competing effect of Umklapp scattering and boundary scattering of heat-carrying phonons at the surface of the material [27]. When the crystal dimensions are smaller or if there are smaller grains in the sample, this will shorten the boundary relaxation time because of the reduction in the mean free path. This means that the boundary scattering process remains dominant up to a

higher temperature, causing a reduction in the thermal conductivity peak, which can be almost undetectable for a small enough grain size. In their study [22], Chen et al. proposed nanostructuring as a possible way to reduce thermal conductivity. Our results seem to suggest that our sample has a natural "nanostructuring" due to our synthesis method. Our thermal conductivity data still exhibit anisotropic results (from oriented grains) but they match favorably with the calculation performed for the grain size of 200 nm in Ref. [22] (based on the values at 300 K). This is consistent with the structure observed in our TEM images (see Figure 4). We note that there have been many efforts to nanostructure HMS samples in the past [28–31].

The two most obvious discrepancies from the DFT phonon dispersion calculation in Chen et al. warrant further discussion. Let us first consider the mode with a 5 meV gap at the zone center in Figure 5d with dispersion given by the red circles in Figure 6. We first note that the position of the 5 meV gap in reciprocal space (1 0 44) is very close to a very strong Bragg peak (1 0 45). In such a case, the resolution function ellipsoid could pick up some of the Bragg intensity and give a false peak in the inelastic scans. However, we checked our experimental configurations (collimation and neutron energy) and confirmed that the resolution function ellipsoid in this case was small enough to not pick up the intensity from the (1 0 45) peak. Since this mode is present near an $(00L)$ Bragg peak with propagation along the $[H00]$ direction, it is most likely a $c$-polarized transverse mode. Keeping this in mind, there are two likely possibilities for this mode. One is that this mode could be a transverse optical (TO) mode, which is not captured in the calculation. However, given the excellent overall agreement between the DFT calculation and the experimental data, this is unlikely. Then, the most likely possibility is that this is the optical mode identified as the twisting mode. In their study, Chen et al. suggested that the twisting mode is a low-lying optical mode that undergoes two successive avoided crossings with the $c$-TA and LA modes at very low energy around 1 meV [22]. Therefore, it is tempting to classify the gap as arising from the avoided crossing between an optical mode of 5 meV and $c$-TA mode. The same avoided crossing may also give rise to the dispersion of the twisting mode.

Next, we will discuss the size of the dispersion, or the bandwidth, of the twisting phonon mode. In the earlier study, the difference between the observed bandwidth of the twisting mode, about 5 meV, and the calculated bandwidth of 7 meV was attributed to the different c-lattice parameters [22]. However, the bandwidth observed in the current study for $Mn_{15}Si_{26}$ is virtually identical to the bandwidth observed by Chen et al. in their study of $Mn_{27}Si_{47}$, despite a difference factor of almost two in the lattice constant. Therefore, the lattice parameter is probably not the main reason for the discrepancy. One possibility is the presence of anharmonic effects, which often show up as the renormalization of phonon frequencies. However, further studies, including temperature dependence and linewidth measurements, will be necessary to address the issue of anharmonicity.

One additional thing to note is that we were not able to observe the twisting mode propagating along the c-direction with a large sound velocity, which was predicted in the DFT calculation but not observed experimentally [22]. Clearly, further investigations with better energy resolutions are required to elucidate the nature of the twisting mode in HMS samples. It will be also interesting to investigate phonon dispersion relations in the $Mn_4Si_7$ phase, which should provide an interesting comparison with the current study and Ref. [22].

## 5. Conclusions

In summary, we synthesized a highly oriented mosaic crystal of $Mn_{15}Si_{26}$ that is large enough to carry out inelastic neutron scattering experiments. We determined the dispersion relation for $Mn_{15}Si_{26}$ to a high degree of completeness along the $[00L]$ and $[H00]$ directions. Our data confirm the experimental and theoretical results reported in earlier neutron scattering and DFT calculations. However, we report two discrepancies from this earlier result. First, we observe an optical mode propagating along the $[H00]$ direction, which has a small gap of 5 meV at the zone center. Second, we found that the lowest-lying mode,

which was assigned as the "twisting" mode, has a smaller energy scale than expected. These results indicate that further work is still needed to fully understand the twisting phonon mode and its role in lattice thermal conductivity of higher manganese silicides.

**Supplementary Materials:** The following supporting information can be downloaded at https://www.mdpi.com/article/10.3390/cryst13050741/s1, Figure S1: Temperature dependence of inelastic neutron scattering data around the (2 0 0) Bragg peak. Panels (a) and (b) are identical to panels (a) and (e) in Figure 5 of the main article, which were obtained at $T = 200$ K. (c) Same as (a) but measured at $T = 10$ K. (d) Same as (b) but measured at $T = 10$ K. One can clearly see the suppression of phonon intensities at lower temperatures, except for the lowest-lying linear branch. This branch originates from the tail of the strong (2 0 0) Bragg peak.

**Author Contributions:** Conceptualization, Y.-C.T. and Y.-J.K.; methodology, R.B., J.P.C., Z.Y., Y.-C.T. and Y.-J.K.; software, R.B.; validation, J.P.C., Z.Y., Y.-C.T. and Y.-J.K.; formal analysis, S.J., R.B.; investigation, R.B.; resources, Y.-C.T. and Y.-J.K.; data curation, R.B., J.P.C. and Z.Y.; writing—original draft preparation, R.B.; writing—review and editing, R.B., J.P.C., Z.Y., Y.-C.T. and Y.-J.K.; visualization, S.J., R.B.; supervision, Y.-J.K.; project administration, Y.-J.K.; funding acquisition, Y.-C.T. and Y.-J.K. All authors have read and agreed to the published version of the manuscript.

**Funding:** Work at the University of Toronto was supported by the Natural Science and Engineering Research Council (NSERC) of Canada through the Collaborative Research and Training Experience (CREATE) program (432242-2013) and a Discovery Grant (RGPIN-2014-06071). Y.-C. Tseng acknowledges the funding provided by Natural Resources Canada through the Program of Energy Research and Development.

**Data Availability Statement:** Data will be made available upon request.

**Acknowledgments:** We acknowledge the Brockhouse Institute for Materials Research, McMaster University, where the powder X-ray diffraction work was carried out.

**Conflicts of Interest:** The authors declare no conflict of interest.

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
