# Peer review of "Inelastic Neutron Scattering Study of Phonon Dispersion Relation in Higher Manganese Silicides"

_crystals, doi:10.3390/cryst13050741_

Round 1

Author Response

We thank the referee for reading our manuscript and providing useful suggestions to improve the manuscript. We have revised the manuscript substantially in response to the referee comments. Most of the figures have been updated, including a major change to Figure 5. We removed the calculation lines from Figure 5 to show our raw data more clearly. This allows to us to provide an in-depth discussion of our inelastic neutron scattering data. In addition, we made numerous changes to the writing and references. Our responses to specific referee comments are provided below. The referee's original comments are italicized.

  1. The authors kept saying that the INS results are mostly consistent with previous studies, as shown in Figure 6. The figure is awful and please relabel the relevant measured data and largely increase the resolution. (All figures need be increased in this manuscript.) It is very necessary to refer to the data processing and plotting of the inelastic scattering experiment in the previous works, including Acta Materialia, 2022, 241: 118390, Nature Communications, 2020, 11(1): 942 and Nature Communications, 2020, 11(1): 3142.

We have updated Fig. 6 as well as other figures.

  1. To justify “what is the international standard unit for thermal conductivity in Figure 3?” and please give the related error bar for measured.

We do not quite understand this comment because thermal conductivity in Figure 3 is presented in SI unit. We, however, included a comment about the error bar.

  1. The author did not explain clearly why the thermal conductivities difference (2-3 times) along in- and out-plane directions. Please add some discussion about it. I would strongly suggest that the authors first carefully revise and improve the quality of this draft, because the current version is more difficult to give a reasonable review comment.

We added discussions about the anisotropy in thermal conductivity.

Reviewer 2 Report

This paper reported the “Inelastic Neutron Scattering Study of Phonon Dispersion

Relation in Higher Manganese Silicides” .This work combined the experimental and calculation works, also, it have enough research content. Therefore, I suggest revision.   

1. We discuss potential origins of these observations, and their implications for the thermal properties of HMS. I think you need to show more details about the origins and properties here.

2. The figure 2 a) need redraw with origin to make it more readable. Also, you need discuss it.

3. I am not very familiar with the Fig. 2 b). I think you need some symbol in the figure to emphasize some part.

4. I suggest you read and add some new works, because you just have 20 ref. such as:10.1016/j.jallcom.2022.167983  and 10.1007/s11467-022-1199-5.

5. Fig 3 need more details and discussion. Eg: why it is higher?

6. Fig 5 a--f, can you changes some colors to make the figure more readable?

7. I suggest you make some DFT calculations, instead just make some ref as Fig. 6.

Author Response

We thank the referee for reading our manuscript and providing useful suggestions to improve the manuscript. We have revised the manuscript substantially in response to the referee comments. Most of the figures have been updated, including a major change to Figure 5. We removed the calculation lines from Figure 5 to show our raw data more clearly. This allows to us to provide an in-depth discussion of our inelastic neutron scattering data. In addition, we made numerous changes to the writing and references. Our responses to specific referee comments are provided below. The referee's original comments are italicized.

  1. “We discuss potential origins of these observations, and their implications for the thermal properties of HMS”. I think you need to show more details about the origins and properties here.

We have added some more discussion about the observed 5 meV gap in the revised draft.

  1. The figure 2 a) need redraw with origin to make it more readable. Also, you need discuss it.

  1. I am not very familiar with the Fig. 2 b). I think you need some symbol in the figure to emphasize some part.

We do not understand what the comment means. The neutron pole figure is extensively discussed in the text. However, we have made changes to the figure to make this more clear.

  1. I suggest you read and add some new works, because you just have 20 ref. such as:10.1016/j.jallcom.2022.167983 and 10.1007/s11467-022-1199-5.

This is a somewhat strange comment. The first suggested reference is already cited in Ref. 18 in our manuscript. The second reference does not seem to be relevant for this current manuscript. However, we did add more references.

  1. Fig 3 need more details and discussion. Eg: why it is higher?

We added discussions about the anisotropy in thermal conductivity.

  1. Fig 5 a--f, can you changes some colors to make the figure more readable?

Figure 5 has been updated.

  1. I suggest you make some DFT calculations, instead just make some ref as Fig. 6.

We would like to emphasize that our paper is an experimental report of phonon dispersion relation, and the comparison with the existing DFT calculation is provided to guide readers understand the different modes observed in our calculation. We do not believe that additional DFT calculation is necessary, since 1) the calculation reported in Ref. 17 is already quite comprehensive and can explain most of our experimentally observed phonon dispersion relation; 2) going beyond the calculation reported in Ref. 17 will require substantially more complex calculation either in terms of unit cell size or the force constant calculations and should be carried out by experts. We note that lattice dynamics calculation is significantly more challenging calculation than the usual band structure calculation.  

Reviewer 3 Report

1st review of "Inelastic Neutron Scattering Study of Phonon Dispersion Relation in Higher Manganese Silicides" by R. Belanger et al.

In this paper the authors present INS study of Mn15Si26. Results are very interesting but few details should be explained before acceptance.

Fig. 5 -probably white lines on some panels are shifted w.r.t. the INS intensity (why at a,b,c for q=0 we haven't 0 frequency modes?)

In Ref. 17 (Nat Commun 6, 6723 (2015)) authors present only Mn4Si7 and Mn27Si47 dispersion relations. Why authors compare INS spectra with Mn4Si7? There is some good argument why this dispersion relations should be comparable? (what with folding? increasing number of ph. branches? and probably decreasing of the lowest ph. modes frequencies?) This all problems are reflected on Fig. 6, where theoretical lines do not reproduce experimental points (right part).

Section Discussion. System is very "non-homogenous" and unit cell is large (c>>a) -- DFT calculations were based on the harmonic phonon calculations. But what with anharmonic effect?  can affect on the thermal conductivity? can you comment this somehow? what another ref tell about this?

Previous study in Ref. 17 (Nat Commun 6, 6723 (2015) - for both Mn4Si7 and Mn27Si47 - fig 2)  show twisiting mode at Gamma with 1 meV, here we have 5 meV - can you explane increasing of this value?

=======

additional remarks:

lines 159, 161, 165 - please use better symbol to "the polarization of phonon modes"

Author Response

We thank the referee for reading our manuscript and providing useful suggestions to improve the manuscript. We have revised the manuscript substantially in response to the referee comments. Most of the figures have been updated, including a major change to Figure 5. We removed the calculation lines from Figure 5 to show our raw data more clearly. This allows to us to provide an in-depth discussion of our inelastic neutron scattering data. In addition, we made numerous changes to the writing and references. Our responses to specific referee comments are provided below. The referee's original comments are italicized.

Fig. 5 -probably white lines on some panels are shifted w.r.t. the INS intensity (why at a,b,c for q=0 we haven't 0 frequency modes?)

Figure 5 has been updated.

In Ref. 17 (Nat Commun 6, 6723 (2015)) authors present only Mn4Si7 and Mn27Si47 dispersion relations. Why authors compare INS spectra with Mn4Si7? There is some good argument why this dispersion relations should be comparable? (what with folding? increasing number of ph. branches? and probably decreasing of the lowest ph. modes frequencies?) This all problems are reflected on Fig. 6, where theoretical lines do not reproduce experimental points (right part).

This is a good point, and we provide a little more discussion about this aspect in the revised version.

Section Discussion. System is very "non-homogenous" and unit cell is large (c>>a) -- DFT calculations were based on the harmonic phonon calculations. But what with anharmonic effect?  can affect on the thermal conductivity? can you comment this somehow? what another ref tell about this?

Anharmonic effect often shows up as renormalization of phonon frequencies. Therefore, the small discrepancies between our data and the DFT calculation results (the low energy ‘twisting’ mode) may point to the significant anharmonic effect. However, further studies including temperature dependence and linewidth measurements will be necessary to address the issue of anharmonicity. We have included the above discussion about the anharmonicity in the revised manuscript.

Previous study in Ref. 17 (Nat Commun 6, 6723 (2015) - for both Mn4Si7 and Mn27Si47 - fig 2)  show twisiting mode at Gamma with 1 meV, here we have 5 meV - can you explane increasing of this value?

We realize that this was not clearly explained in the original submission. The twisting mode at Gamma was “predicted” to be at 1 meV, but the coarse energy resolution prevented the authors from observing this at the Gamma point in Ref. 17. Here, we observe a mode at 5 meV at the Gamma point, but we are not sure about the exact nature of this gap. This mode can be the twisting mode, but again, a further studies using a better resolution will be required to confirm this.

additional remarks:

lines 159, 161, 165 - please use better symbol to "the polarization of phonon modes"

We apologize for this. This was a typesetting error, and we corrected it.

Round 2

Reviewer 1 Report

Overall, I am satisfied with the changes,regarding the data and figures described, and the authors updated it. However, the more detailed crystal structure parameters for Mn15Si26 is still confusing (having space group and   structural disorder induced the anisotropic Kappa). I would like to recommend it for publication once the problem has been addressed.

Author Response

We thank the reviewer for reading our revised manuscript and providing feedback. However, we do not understand the comment by the reviewer: "However, the more detailed crystal structure parameters for Mn15Si26 is still confusing (having space group and   structural disorder induced the anisotropic Kappa)." In order to address this comment, we would appreciate if the reviewer can elaborate on what the confusing part is. 

Reviewer 2 Report

I have no comment about this paper. It can be accepted.

Author Response

We would like to thank the reviewer for reading our revised manuscript. This manuscript was written by native English speakers and we do not find any major issues with English. If there is an error found by the reviewer, we would be happy to correct it.